# Phytochemical Investigation of *Tradescantia Albiflora* and Anti-Inflammatory Butenolide Derivatives

**DOI:** 10.3390/molecules24183336

**Published:** 2019-09-13

**Authors:** Ping-Chen Tu, Han-Chun Tseng, Yu-Chia Liang, Guan-Jhong Huang, Te-Ling Lu, Tzong-Fu Kuo, Yueh-Hsiung Kuo

**Affiliations:** 1The Ph.D. Program for Cancer Biology and Drug Discovery, China Medical University and Academia Sinica, Taichung 404, Taiwan; pingchen.tu@gmail.com; 2Department of Chemistry, National Taiwan University, Taipei 106, Taiwan; hcts97@gmail.com; 3Department of Chinese Pharmaceutical Sciences and Chinese Medicine Resources, China Medical University, Taichung 404, Taiwan; allen1987323@yahoo.com.tw (Y.-C.L.);; 4School of Pharmacy, China Medical University, Taichung 404, Taiwan; lutl@mail.cmu.edu.tw; 5Department of Post-Baccalaureate Veterinary Medicine, Asia University, Taichung 413, Taiwan; tzongfu@asia.edu.tw; 6Department of Biotechnology, Asia University, Taichung 413, Taiwan; 7Chinese Medicine Research Center, China Medical University, Taichung 404, Taiwan

**Keywords:** *Tradescantia albiflora*, butenolides, anti-inflammatory activity

## Abstract

Phytochemical investigation of the whole plant of *Tradescantia albiflora* Kunth led to the isolation and characterization of a butanolide, rosmarinosin B (**1**), that was isolated from natural sources for the first time, a new butenolide, 5-*O*-acetyl bracteanolide A (**2**), and a new apocarotenoid, 2*β*-hydroxyisololiolide (**11**), together with 25 known compounds (compounds **3**–**10** and **12**–**28**). The structures of the new compounds were elucidated by analysis of their spectroscopic data, including MS, 1D, and 2D NMR experiments, and comparison with literature data of known compounds. Furthermore, four butenolides **4a**–**4d** were synthesized as novel derivatives of bracteanolide A. The isolates and the synthesized derivatives were evaluated for their preliminary anti-inflammatory activity against lipopolysaccharide (LPS)-stimulated nitric oxide (NO) production in RAW 264.7 cells. Among them, the synthesized butenolide derivative *n*-butyl bracteanolide A (**4d**) showed enhanced NO inhibitory activity compared to the original compound, with an IC_50_ value of 4.32 ± 0.09 μg/mL.

## 1. Introduction

*Tradescantia albiflora* Kunth (Commelinaceae) is native to tropical rainforests. It has been used as a traditional medicine for treating hyperuricemia and gout in Taiwan. Previous research described the inhibitory activity against xanthine oxidase (XO), which plays a central role in metabolic disorders such as hyperuricemia and gout, of the methanol extract and compounds isolated from the leaves of *T. albiflora* [1]. However, none of the isolated compounds showed significant inhibitory activity.

Our continuing investigation on the bioactive compounds from *T. albiflora* has now led to the extraction, purification, and structural elucidation of three new naturally occurring compounds, together with 25 known compounds. The butenolide bracteanolide A (**4**) was the most abundant compound among the isolates, and it has been reported to show inhibitory ability against lipopolysaccharide (LPS)-stimulated nitric oxide (NO) production in RAW 264.7 cells. This inhibition was associated with its selective suppression on inducible NO synthase (iNOS) induction [2], indicating its potential to treat inflammatory diseases caused by NO production. For this reason, bracteanolide A (**4**) was used as a starting material for the preparation of butenolide derivatives. In addition, the isolates and four newly synthesized derivatives were evaluated for their preliminary anti-inflammatory activity against LPS-stimulated NO production in RAW 264.7 cells.

## 2. Results and Discussion

Phytochemical investigation of the whole plants of *T. albiflora* Kunth led to the isolation and characterization of three new compounds and 25 known compounds, which were identified by comparison with literature spectroscopic data and determined as 4-(3’,4’-dihydroxyphenyl)furan-2(5*H*)-one (**3**) [3], bracteanolide A (**4**) [2], bracteanolide B (**5**) [2], methyl 3,4-dihydroxybenzoate (**6**) [4], hydroxytyrosol (**7**) [5], 1-(3,4-dihydroxyphenyl)-2-hydroxyethan-1-one (**8**) [5], (±)-tradescantin (**9**) [3], tricin (**10**) [6], isololiolide (**12**) [7], loliolide (**13**) [8], (3*R*)-3-hydroxy-*β*-ionone (**14**) [9], (6*R*,7*E*,9*R*)-9-hydroxy-4,7-megastigmadien-3-one (**15**) [10], (*E*)-3,5,5-trimethyl-4-(3-oxobut-1-en-1-yl)cyclohex-2-enone (**16**) [11], (*S*)-dehydrovomifoliol (**17**) [12], *N*-*trans*-feruloyltyramine (**18**) [13], *N*-*trans*-feruloyl-3-methoxytyramine (**19**) [13], sitosterol (**20**) [14], stigmasterol (**21**) [14], 7-ketositosterol (**22**) [15], 7-ketostigmasterol (**23**) [15], 7*β*-hydroxysitosterol (**24**) [15], schottenol (**25**) [14], ergosterol peroxide (**26**) [16], 24,25-dihydrocimicifugenol (**27**) [17], 3-epicyclomusalenol (**28**) [17] (Figure 1).

Compound **1** was obtained as a colorless amorphous solid with ${\left[\alpha \right]}_{D}^{25}$ − 13.5, and its high resolution electrospray ionization mass spectrometry (HRESIMS) data determined the molecular formula as C_10_H_10_O_4_ (*m*/*z* 217.0464, assigned as C_10_H_10_O_4_Na) indicating six degrees of unsaturation. The IR spectrum displayed the presence of hydroxyl (3312 cm^−1^), *γ*-lactone (1758 cm^−1^), and aromatic (1607, 1526 cm^−1^) functionalities.

The ^1^H-NMR spectrum (Table 1) displayed signals characteristic of a trisubstituted benzene ring indicated by an ABX-pattern for three aromatic protons [*δ*_H_ 6.73 (d, *J* = 8.4 Hz), 6.61 (dd, *J* = 8.4, 2.0 Hz), and 6.71 (d, *J* = 2.0 Hz)] and a butanolide moiety deduced from the following spectroscopic data: one pair of oxymethylene protons [*δ*_H_ 4.62 (t, *J* = 8.0 Hz) and 4.20 (t, *J* = 8.0 Hz)], one pair of lactone methylene protons [*δ*_H_ 2.86 (dd, *J* = 17.4, 8.6 Hz) and 2.62 (dd, *J* = 17.4, 8.9 Hz)], and one methine proton (*δ*_H_ 3.68, m), together with the IR peak at *ν*_max_ 1758 cm^−1^. The HMBC correlations from H-3a, H-3b, H-4, H-5a, and H-5b to C-1’ (*δ*_C_ 133.0) indicated that the butanolide functionality was attached on C-1’. This assignment was confirmed by the deshielded signal of benzylic proton at *δ*_H_ 3.68 (H-4). According to the overall specific rotation, 1D, and 2D NMR, compound **1** was determined as rosmarinosin B with a 3*R* configuration, which was previously obtained as an artificial compound by gamma irradiation-assisted degradation of rosmarinic acid and exhibited moderately enhanced anti-adipogenic properties in 3T3-L1 cells than the original compound [18]. It was isolated from the natural sources for the first time.

Compound **2** was obtained as a colorless amorphous solid with ${\left[\alpha \right]}_{D}^{25}$ + 2.5, and its HRESIMS data determined the molecular formula as C_12_H_10_O_6_ (*m*/*z* 249.0371, assigned as C_12_H_9_O_6_), indicating eight degrees of unsaturation. The IR spectrum displayed the presence of hydroxyl (3470, 3169 cm^−1^), *γ*-lactone (1757 cm^−1^), ester (1728 cm^−1^), and aromatic (1609, 1516 cm^−1^) functionalities.

The ^1^H-NMR spectrum (Table 1) displayed signals characteristic of the presence of two hydroxyl groups attached on the benzene ring (*δ*_H_ 8.63, brs) as determined by D_2_O exchange experiment, a highly deshielded oxymethine (*δ*_H_ 7.40, s, H-5), a trisubstituted benzene ring indicated by an ABX-pattern for three aromatic protons [*δ*_H_ 7.15 (d, *J* = 2.1 Hz), 7.11 (dd, *J* = 8.3, 2.1 Hz), and 6.94 (d, *J* = 8.3 Hz)], a conjugated olefinic proton (*δ*_H_ 6.49, s, H-3), and an acetyl group (*δ*_H_ 2.15, 3H, s). Additionally, twelve carbon signals were displayed in the ^13^C-NMR spectrum. The assignments of two *ortho*-hydroxyl groups attached on the benzene ring were confirmed by two deshielded signals of aromatic carbons at *δ*_C_ 150.7 and 146.8. The IR peak at *ν*_max_ 1757 cm^−1^ for *γ*-lactone functionality together with the carbon signals at *δ*_C_ 171.0, 162.8, 112.8, and 93.5 indicated the presence of the oxygenated unsaturated butenolide moiety. This moiety was also deduced from the HSQC correlations from H-3 (*δ*_H_ 6.49 s) to C-3 (*δ*_C_ 112.8) and from H-5 (*δ*_H_ 7.40 s) to C-5 (*δ*_C_ 93.5). The HMBC correlation from H-5 (*δ*_H_ 7.40, s) to the acetyl carbon (*δ*_C_ 170.0) indicated that the acetoxyl group was attached on the C-5 in butenolide moiety. The overall 1D and 2D NMR data suggested the structural similarities between **2** and bracteanolide A (**4**), except that the hydroxy group on C-5 in **4** was replaced by the *O*-acetyl group. This assignment was also confirmed by a highly deshielded oxymethine signal (*δ*_H_/*δ*_C_ 7.40/93.5). Compound **2** was consequently determined as new butenolide, named 5-*O*-acetyl bracteanolide A.

Compound **11** was obtained as a colorless oil with ${\left[\alpha \right]}_{D}^{25}$ + 27.6, and its high resolution electron ionization mass spectrometry (HREIMS) peak at *m*/*z* 212.1043 determined the molecular formula as C_11_H_16_O_4_, indicating four degrees of unsaturation. The IR spectrum displayed the presence of hydroxyl (3406 cm^−1^) and lactone (1741 cm^−1^) functionalities.

The ^1^H-NMR spectrum (Table 2) indicated the presence of an olefinic proton signal at *δ*_H_ 5.78 (1H, s), two oxymethine signals at *δ*_H_ 3.80 and 3.04, a methylene signal at *δ*_H_ 2.38 and 1.43, and three methyl signals at *δ*_H_ 1.58, 1.33, and 1.18 (each 3H, s). ^13^C-NMR and DEPT experiments revealed the presence of **11** carbon signals, indicating a lactone carbon at *δ*_C_ 170.7, a pair of conjugated carbon at *δ*_C_ 180.3 and 114.0, one oxygenated quaternary carbon at *δ*_C_ 85.3, two oxymethines at *δ*_C_ 81.7 and 67.9, a methylene at *δ*_C_ 43.6, and three methyls at *δ*_C_ 25.7, 25.2, and 18.7.

An additional hydroxy group was assigned at C-2 (*δ*_C_ 81.7) of **11** by comparing the NMR data of **12**. The planar structure of **11** was confirmed by HMBC correlations shown in Figure 2. The di-axial orientations of H-2 and H-3 were deduced from the coupling constant of 9.3 Hz between them. The NOESY correlations between H-2 and Me-9; H-3 and Me-11; Me-10 and Me-11 established the relative configuration of **11**. Thus, compound **11** was determined as a new apocarotenoid, named 2*β*-hydroxyepiloliolide.

In this study, the phytochemical investigation on the bioactive compounds from *T. albiflora* led to the isolation of 28 compounds including a butanolide rosmarinosin B (**1**) and four butenolides, 5-*O*-acetyl bracteanolide A (**2**), 4-(3’,4’-dihydroxyphenyl)furan-2(5*H*)-one (**3**), bracteanolide A (**4**), and bracteanolide B (**5**). Among the isolates, the butanolide bracteanolide A (**4**) was highly abundant in this plant (1.31 mg/g extract). It has been reported to exhibit inhibitory activity against LPS-activated NO production in RAW 264.7 cells by suppressing iNOS expression selectively [2], which is the potential target for the treatment of the inflammatory diseases caused by NO production. In order to screen the naturally anti-inflammatory butenolides and their related derivatives, four new butenolide derivatives **4a**–**4d** were synthesized by modification from bracteanolide A (**4**) at C-4, focusing on changing the hydroxy group into alkoxy groups (Scheme 1). Then, the isolated compounds **2**–**8** and four new butenolide derivatives **4a**–**4d** were evaluated for their preliminary anti-inflammatory activity against NO production in RAW 264.7 cells, a reliable indicator in investigating inflammatory activity [19].

All compounds evaluated displayed lower anti-inflammatory activity than the positive control dexamethasone (Table 3), which has been reported to decrease iNOS-dependent NO production [20]. Dexamethasone is a highly effective anti-inflammatory and immunosuppressant corticosteroid. Unfortunately, the long-term use may cause serious systemic side effects, ranging from weight gain, diabetes, hypertension, immunosuppression, psychological disturbances, fragile skin, muscle weakness, osteoporosis, and Cushing’s syndrome [21].

As shown in Table 3 and Appendix A, compounds **4**, **4b**, **4d**, **6**, and **7** showed inhibitory potential against NO production, which was not associated with their cytotoxicity against RAW 264.7 cells (Appendix A). The results suggested that the disappearance of the hydroxy group (**3**) or the presence of methyl (**5**), ethyl (**4a**), *i*-propyl (**4c**), and acetyl groups (**2**) resulted in a decreased activity; the presence of *n*-propyl group did not affect the activity. The data also revealed that the catechol group might have contributed to the activity even if the result of compound **8** was unsatisfactory (the IC_50_ value was above 50 µg/mL). Among compounds evaluated, compound **4d** with an *n*-butyl group showed enhanced anti-inflammatory activity (IC_50_ value of 4.32 ± 0.09 μg/mL) compared to the original compound (Table 3).

## 3. Materials and Methods

### 3.1. General

Optical rotations were measured with a DIP-1000 Polarimeter (JASCO, Tokyo, Japan). UV spectra were recorded on a Heλios Beta UV-Visible spectrophotometer (Thermo Scientific Inc., Waltham, MA, USA). Infrared spectra were acquired on a Nicolet MAGNA-IR 500 spectrophotometer (Thermo Scientific Inc., Waltham, MA, USA). The NMR experiments were performed on DMX-400 and DMX-500 MHz NMR spectrometers (Bruker, Bremen, Germany). HREIMS and HRESIMS spectra were generated with SX-102A (JEOL, Tokyo, Japan) and maXis impact mass spectrometers (Bruker Daltonics, Bremen, Germany), respectively. Column chromatography was performed on Silica gel 60 (40–63 µm, Merck, Darmstadt, Germany), high performance liquid chromatography (HPLC) was performed using Keystone Spherisorb silica (5 µm, 250 × 10 mm), and thin-layer chromatography (TLC) was performed on silica gel 60 F254 plates (200 µm, Merck).

### 3.2. Plant Material

The greenhouse-grown plant material was obtained from Dr. T.-F.K. (Department of Post-Baccalaureate Veterinary Medicine, Asia University). A voucher specimen (TAIF-PLANT-199332) has been retained at the Herbarium of Taiwan Forestry Research Institute, Taipei, Taiwan.

### 3.3. Extraction and Isolation

Air-dried whole plant of *T. albiflora* (14.9 kg) was extracted twice with methanol (40 L) at room temperature for 7 days, and concentrated under vacuum. The methanol extract (1.5 kg) taken up in distilled water was fractionated successively with ethyl acetate and *n*-butanol to yield the corresponding solvent-soluble fractions. The ethyl acetate-soluble fraction (232.7 g) was subjected to silica gel column chromatography (2.0 kg, 70–230 mesh) using a gradient solvent system (*n*-hexane/ethyl acetate/methanol) as eluant to afford 10 fractions. Fr. 3 (35.2 g) was reseparated by silica gel column chromatography (*n*-hexane/acetone = 95/5) followed by semi-preparative normal phase HPLC to give compounds **25** (10.7 mg), **27** (7.6 mg), and **28** (7.4 mg). Fr. 4 (24.6 g) was reseparated by silica gel column chromatography (dichloromethane/acetone = 95/5) and recrystallization to obtain compounds **20**/**21** (740.2 mg) and **26** (8.8 mg). Fr. 5 (9.0 g) was reseparated by silica gel column chromatography using a gradient solvent system (dichloromethane/ethyl acetate) to afford 10 fractions. Fr. 5-2 was purified by normal phase HPLC (dichloromethane/acetone = 80/20) to give compounds **10** (8.1 mg), **12** (57.9 mg), **16** (9.0 mg), and **17** (11.0 mg). Fr. 5-4 was purified by normal phase HPLC (dichloromethane/acetone = 80/20) to obtain compounds **11** (4.7 mg), **14** (13.0 mg), and **22**/**23** (44.1 mg). Fr. 5-5 was purified by normal phase HPLC (*n*-hexane/acetone = 80/20) to obtain compound **24** (13.6 mg). Fr. 5-7 purified by normal phase HPLC (dichloromethane/ethyl acetate = 65/35) to obtain compound **9** (27.4 mg). Fr. 6 (18.4 g) was reseparated by silica gel column chromatography a gradient solvent system (dichloromethane/ethyl acetate) to afford 12 fractions. Fr. 6-2 was purified by normal phase HPLC (dichloromethane/ethyl acetate = 67/33) to obtain compounds **6** (8.8 mg) and **13** (30.1 mg). Fr. 6-3 was purified by normal phase HPLC (dichloromethane/ethyl acetate = 50/50) to obtain compound **15** (9.0 mg). Fr. 6-6 was purified by normal phase HPLC (*n*-hexane/acetone = 75/25) to obtain compounds **1** (5.1 mg) and **2** (10.3 mg). Fr. 6-7 was purified by normal phase HPLC (*n*-hexane/ethyl acetate = 30/70) to obtain compounds **3** (8.8 mg), **4** (1958.6 mg), and **5** (30.1 mg). Fr. 6-8 was purified by normal phase HPLC (dichloromethane/acetone = 75/25) to obtain compounds **7** (13.1 mg) and **8** (8.0 mg). Fr. 6-8 was purified by normal phase HPLC (dichloromethane/acetone = 70/30) to obtain compounds **18** (8.9 mg) and **19** (7.1 mg).

*Rosmarinosin B* (**1**): Colorless amorphous solid; ${\left[\alpha \right]}_{D}^{25}$ − 13.5 (c 0.1, MeOH); UV (MeOH) *λ*_max_ (log ε): 283 (2.54) nm; IR (KBr) *ν*_max_ 3312, 2918, 2851, 1759, 1607, 1526, 1449, 1375, 1285, 1188, 1117, and 1015 cm^−1^; HRESIMS *m*/*z* 217.0464 [M + Na]^+^ (C_10_H_10_O_4_Na); ^1^H- and ^13^C-NMR: see Table 1.

*5-**O**-Acetylbracteanolide* A (**2**): Colorless amorphous solid ${\left[\alpha \right]}_{D}^{25}$ + 2.5 (c 0.1, MeOH); UV (MeOH) *λ*_max_ (log ε): 217 (4.08), 249 (4.02), 334.6 (4.19) nm; IR (KBr) *ν*_max_ 3470, 3169, 2961, 2924, 1757, 1728, 1609, 1516, 1302, 1285, 1223, 1198, 1177, 1032, and 989 cm^−1^; HRESIMS *m*/*z* 249.0371 [M − H]^−^ (C_12_H_9_O_6_); ^1^H- and ^13^C-NMR: see Table 1.

*2**β**-Hydroxyisololiolide* (**11**): Colorless oil; ${\left[\alpha \right]}_{D}^{25}$ + 27.6 (c 0.27, MeOH); UV (MeOH) *λ*_max_ (log ε): 212 (3.94), 272 (2.79) nm; IR (KBr) *ν*_max_ 3406, 2925, 2873, 1742, 1629, 1460, 1383, 1291, 1260, 1050, 984, and 938 cm^−1^; HREIMS *m*/*z* 212.1043 [M]^+^ (C_11_H_16_O_4_); ^1^H- and ^13^C-NMR: see Table 2.

### 3.4. Preparation of Butenolide Derivatives ***4a–4d***

Bracteanolide A (20 mg) was dissolved in the corresponding alcohol (10 mL). Concentrated hydrochloric acid (3 drops) was added and the solution was refluxed and stirred for 12 h. The solvent was evaporated under vacuum to produce a yellow residue that was diluted with distilled water and fractionated twice with dichloromethane. The organic extracts were combined and dried over magnesium sulfate to give butenolide derivatives **4a**–**4d**, in approximately 70% yield.

*5-**O**-Ethyl bracteanolide A* (**4a**): Yield 73%; IR (KBr) *ν*_max_ 3368, 3098, 2974, 1728, 1605, 1512, 1373, 1342, 1300, 1281, 1200, 1177, 1115, 1015, 972, and 945 cm^−1^; ^1^H-NMR (acetone-*d*_6_, 400 MHz): *δ*_H_ 7.28 (d, 1H, *J* = 2.1 Hz), 7.21 (dd, 1H, *J* = 8.3, 2.1 Hz), 6.92 (d, 1H, *J* = 8.3 Hz), 6.38 (s, 1H), 6.36 (s, 1H), 3.84 (m, 2H), 1.24 (t, 3H, *J* = 7.1 Hz); ^13^C-NMR: *δ*_C_ 170.5, 161.7, 148.8, 145.4, 121.7, 120.9, 115.5, 114.7, 112.2, 102.1, 64.2, 14.5.

*5-**O**-**n**-**Propyl**bracteanolide A* (**4b**): Yield 70%; IR (KBr) *ν*_max_ 3472, 3167, 2970, 2878, 1713, 1605, 1512, 1408, 1342, 1300, 1285, 1200, 1177, 1123, 1030, 964, and 949 cm^−1^; ^1^H-NMR (acetone-*d*_6_, 400 MHz): *δ*_H_ 7.28 (d, 1H, *J* = 1.9 Hz), 7.21 (dd, 1H, *J* = 8.3, 1.9 Hz), 6.93 (d, 1H, *J* = 8.3 Hz), 6.38 (s, 1H), 6.37 (s, 1H), 3.77 (t, 2H, *J* = 7.1 Hz), 1.63 (tq, 2H, *J* = 7.0 Hz), 0.92 (t, 3H, *J* = 7.0 Hz); ^13^C NMR: *δ*_C_ 170.5, 161.7, 148.7, 145.4, 121.7, 121.0, 115.5, 114.8, 112.2, 102.3, 70.1, 22.6, 9.9.

*5-**O**-**i**-**Propyl**bracteanolide A* (**4c**): Yield 67%; IR (KBr) *ν*_max_ 3472, 3159, 2978, 2920, 2808, 1713, 1605, 1512, 1408, 1381, 1323, 1342, 1300, 1281, 1196, 1150, 1126, 1030, 961, and 922 cm^−1^; ^1^H-NMR (acetone-*d*_6_, 400 MHz): *δ*_H_ 7.26 (d, 1H, *J* = 2.1 Hz), 7.19 (dd, 1H, *J* = 8.3, 2.1 Hz), 6.92 (d, 1H, *J* = 8.3 Hz), 6.44 (s, 1H), 6.33 (s, 1H), 4.22 (m, 2H), 1.33 (t, 3H, *J* = 6.1 Hz), 1.23 (t, 3H, *J* = 6.1 Hz); ^13^C NMR: *δ*_C_ 170.6, 162.1, 148.7, 145.3, 121.8, 120.9, 115.5, 114.8, 112.1, 101.4, 72.8, 22.8, 21.6.

*5-**O**-**n**-**Butyl**bracteanolide A* (**4d**): Yield 69%; IR (KBr) *ν*_max_ 3476, 3171, 2963, 2936, 2874, 1732, 1605, 1512, 1412, 1373, 1342, 1300, 1200, 1126, 1030, 972, 941, and 929 cm^−1^; ^1^H-NMR (acetone-*d*_6_, 400 MHz): *δ*_H_ 7.27 (d, 1H, *J* = 1.9 Hz), 7.21 (dd, 1H, *J* = 8.3, 1.9 Hz), 6.92 (d, 1H, *J* = 8.3 Hz), 6.38 (s, 1H), 6.36 (s, 1H), 3.78 (m, 2H), 1.60 (dq, 2H, *J* = 6.6 Hz), 1.37 (tq, 2H, *J* = 6.2 Hz), 0.89 (t, 3H, *J* = 7.4 Hz); ^13^C NMR: *δ*_C_ 170.5, 161.7, 148.8, 145.4, 121.6, 120.9, 115.5, 114.7, 112.2, 102.3, 68.1, 31.4, 18.9, 13.1.

### 3.5. Cell Culture

A murine macrophage cell line RAW264.7 (BCRC No. 60001) was obtained from the Bioresources Collection and Research Center of the Food Industry Research and Development Institute (Hsinchu, Taiwan). Cells were maintained in Dulbecco′s Modified Eagle Medium (DMEM, Sigma, St. Louis, MO, USA) supplemented with 10% fetal bovine serum (FBS, Sigma) in an incubator containing 5% CO_2_ at 37 °C and subcultured every 3 days using 0.05% trypsin-0.02% EDTA in Ca^2+^-, Mg^2+^-free phosphate-buffered saline (DPBS).

### 3.6. Cell Viability

Raw 264.7 cells (5 × 10^4^ cells/well) were seeded into 96-well plates and incubated for 24 h. Then, cells were treated with different concentrations of samples in the presence of 100 ng/mL LPS. After incubation overnight, the cells were washed twice with DPBS and incubated with 100 μL MTT (0.5 mg/mL) for 3 h. The medium was removed, and MTT formazan was dissolved by 100 μL dimethyl sulfoxide (DMSO). Then, absorbance at 570 nm was read using a microplate reader.

### 3.7. Measurement of Nitric Oxide/Nitrite

NO production was indirectly measured by determining the nitrite levels in the cultured medium using a colorimetric assay based on the Griess reaction. Cells were treated with different concentrations of samples in the presence of LPS (100 ng/mL) and incubated for 24 h. Then, each supernatant (100 μL) was mixed with the same volume of Griess reagent (1% sulfanilamide, 0.1% naphthylethylenediamine dihydrochloride and 5% phosphoric acid) and incubated for 5 min, and the absorbance at 540 nm was measured using a microplate reader.

## 4. Conclusions

In summary, phytochemical investigation of the whole plant of *Tradescantia albiflora* Kunth has led to the isolation and characterization of a butanolide, rosmarinosin B (**1**), that was isolated from natural sources for the first time, a new butenolide, 5-*O*-acetyl bracteanolide A (**2**), and a new apocarotenoid 2*β*-hydroxyisololiolide (**11**), together with 25 known compounds (compounds **3**–**10** and **12**–**28**). The isolated compounds **2**–**8** and four new synthetic butenolide derivatives **4a**–**4d**, which were synthesized from bracteanolide A (**4**), were evaluated for their preliminary anti-inflammatory activity against LPS-stimulated NO production in RAW 264.7 cells. Among them, the new synthetic butenolide derivative *n*-butyl bracteanolide A (**4d**) exhibited better NO inhibitory activity than the original compound bracteanolide A (**4**).

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
