# Peer review of "Phytochemical Investigation of Tradescantia Albiflora and Anti-Inflammatory Butenolide Derivatives"

_molecules, 2019, doi:10.3390/molecules24183336_

Round 1
Reviewer 1 Report
The manuscript written by Tu et al. describes the identification of a couple of new plant metabolites from Tradescantia albiflora. In addition, they evaluated the anti-inflammatory effect of the compounds isolated from T. albiflora. The work looks interesting; however, they need additional information on biology area.
The method for MTT assay is written; however, there are no data on MTT assay.
Why did the authors have an interest in compound 4 and synthesize its analogues? Please explain the reason.
Please add discussion about the structure-activity relationship between the compounds used and the anti-inflammatory activity.
Author Response
Point 1: The method for MTT assay is written; however, there are no data on MTT assay.
Response 1: Figure S23 was supplemented in Supplementary Material. The cytotoxicity on RAW 264.7 cells was also described in the manuscript.
Point 2: Why did the authors have an interest in compound 4 and synthesize its analogues? Please explain the reason.
Response 2: The butanolide bracteanolide A (4) was highly abundant among the isolated compounds in this plant (1.31 mg/g extract). It has been reported to exhibit NO inhibitory activity in LPS-activated RAW 264.7 cells by suppressing iNOS expression selectively. Thus, it could act as a starting material for synthesized derivatives for bioactive assessment. Then, the isolated compounds and synthesized butenolide derivatives from bracteanolide A were evaluated for their NO inhibitory activity in LPS-activated RAW 264.7 cells (shown in line 123-135).
Point 3: Please add discussion about the structure-activity relationship between the compounds used and the anti-inflammatory activity.
Response 3: The discussion about the structure-activity relationship between the compounds was added in the manuscript (shown in line137-151).
Reviewer 2 Report
Dear, Yueh-Hsiung Kuo, Ph.D.
Please check and revise.
-------------------------------------------------------------------------------------------------------------------------
Compound 11: A HMBC correlation from H-11 to C-4 is necessary for structural elucidation.
Compound 2: I think this compound is artificial. You used EtOAc with Extraction procedure. And I think this compound is unintentionally a same artificial compound as 4a-4d.
Anti-inflammatory activity assay: Why were not 1 and 11 evaluated? These compounds were enough amounts for assay.
-------------------------------------------------------------------------------------------------------------------------
Author Response
Point 1: Compound 11: A HMBC correlation from H-11 to C-4 is necessary for structural elucidation.
Response 1: Figure 2. was revised.
Point 2: Compound 2: I think this compound is artificial. You used EtOAc with Extraction procedure. And I think this compound is unintentionally a same artificial compound as 4a-4d.
Response 2: Acyl halides or acetic anhydride, not available during extraction and purification, are necessary for the reaction to synthesize 2. Therefore, the compound 2 could be supposed not to be artificial.
Point 3: Anti-inflammatory activity assay: Why were not 1 and 11 evaluated? These compounds were enough amounts for assay.
Response 3: Actually, we don’t have the sample (11) for investigation now. After a period of time, compound 1 was found to be less stable than other compounds. The authors tried to purify and elucidate the reaction. However, the final product was still unknown.
Reviewer 3 Report
Some suggestion:
1. The activites of the compounds is not as potent as the positive control and I think ot is not appropriate to conclude them as potent anti-inflammatory activity in the manuscript.
2. For RAW cells study, the authors may also measure the activites of the compounds on TNFa expression, as NO alone is not too convincing for anti-inflammatory activity of those compounds in the whole picture.
3. Cytotoxicity of the compounds on different cells should be added to demonstrate their anti-inflammatory activities are not toxic.
Author Response
Point 1: The activites of the compounds is not as potent as the positive control and I think ot is not appropriate to conclude them as potent anti-inflammatory activity in the manuscript.
Response 1: The description was revised. The adverse effects of the positive control were also described.
Point 2: For RAW cells study, the authors may also measure the activites of the compounds on TNFa expression, as NO alone is not too convincing for anti-inflammatory activity of those compounds in the whole picture.
Response 2: The butanolide bracteanolide A, which was used as a starting material for derivatives, was highly abundant in this plant (1.31 mg/g extract). It has been reported to exhibit inhibitory activity against LPS-activated NO production in RAW 264.7 cells by suppressing iNOS expression selectively. Thus, we assessed the NO inhibitory activity of these isolated and synthesized compounds. The results were preliminary. Further study demands more detailed investigations. NO alone is not too convincing but still one of the reliable indicators in investigating inflammatory activity. (The published articles in Molecules: Yang et al., 2017 and Zhang et al., 2019)
Point 3: Cytotoxicity of the compounds on different cells should be added to demonstrate their anti-inflammatory activities are not toxic.
Response 3: Figure S23 was supplemented in Supplementary Material. The cytotoxicity on RAW 264.7 cells was also described in the manuscript. The results suggested that inhibitory potential of the samples against LPS-activated NO production, which was not associated with their cytotoxicity.
Round 2
Reviewer 3 Report
No further comments.